# The molecular mechanism for carbon catabolite repression of the chitin response in *Vibrio cholerae*

**Virginia E. Green**[1], **Catherine A. Klancher**[1], **Shouji Yamamoto**[2], **Ankur B. Dalia**[1]*

**1** Department of Biology, Indiana University, Bloomington, Indiana, United States of America, **2** Department of Bacteriology I, National Institute of Infectious Diseases, Tokyo, Japan

* ankdalia@indiana.edu

**Data Availability Statement:** All relevant data are within the manuscript and its Supporting Information files.

**Funding:** This work was supported by grant R35GM128674 from the National Institutes of

## Abstract

*Vibrio cholerae* is a facultative pathogen that primarily occupies marine environments. In this niche, *V. cholerae* commonly interacts with the chitinous shells of crustacean zooplankton. As a chitinolytic microbe, *V. cholerae* degrades insoluble chitin into soluble oligosaccharides. Chitin oligosaccharides serve as both a nutrient source and an environmental cue that induces a strong transcriptional response in *V. cholerae*. Namely, these oligosaccharides induce the chitin sensor, ChiS, to activate the genes required for chitin utilization and horizontal gene transfer by natural transformation. Thus, interactions with chitin impact the survival of *V. cholerae* in marine environments. Chitin is a complex carbon source for *V. cholerae* to degrade and consume, and the presence of more energetically favorable carbon sources can inhibit chitin utilization. This phenomenon, known as carbon catabolite repression (CCR), is mediated by the glucose-specific Enzyme IIA (EIIA$^{Glc}$) of the phosphoenolpyruvate-dependent phosphotransferase system (PTS). In the presence of glucose, EIIA$^{Glc}$ becomes dephosphorylated, which inhibits ChiS transcriptional activity by an unknown mechanism. Here, we show that dephosphorylated EIIA$^{Glc}$ interacts with ChiS. We also isolate ChiS suppressor mutants that evade EIIA$^{Glc}$-dependent repression and demonstrate that these alleles no longer interact with EIIA$^{Glc}$. These findings suggest that EIIA$^{Glc}$ must interact with ChiS to exert its repressive effect. Importantly, the ChiS suppressor mutations we isolated also relieve repression of chitin utilization and natural transformation by EIIA$^{Glc}$, suggesting that CCR of these behaviors is primarily regulated through ChiS. Together, our results reveal how nutrient conditions impact the fitness of an important human pathogen in its environmental reservoir.

## Author summary

*Vibrio cholerae* is a facultative human pathogen that most commonly inhabits aquatic environments and can cause the disease cholera when ingested in the form of contaminated food or drinking water. The survival of this pathogen in the marine environment is dependent upon its ability to colonize chitin, an insoluble polysaccharide that is the main

Health to ABD. The funders had no role in study design, data collection and analysis, decision to publish, or preparation of the manuscript.

**Competing interests:** The authors have declared that no competing interests exist.

constituent of the shells of crustacean zooplankton. *V. cholerae* degrades chitin into soluble oligomers, which can be taken up by this microbe and used as a nutrient. Importantly, chitin oligomers also induce the activity of a chitin-sensing protein, ChiS, to activate the expression of genes required for both (1) eating chitin and (2) undergoing horizontal gene transfer by natural transformation. Previous work demonstrated that more energetically favorable carbon sources, like glucose, can inhibit chitin-induced behaviors. However, the mechanism underlying this response remained unclear. Here, we show how a dephosphorylated intermediate in the glucose uptake pathway–EIIA$^{Glc}$–interacts with ChiS to repress its activity. Furthermore, we demonstrate that this repression of ChiS by EIIA$^{Glc}$ is responsible for inhibiting both growth on chitin and natural transformation. Together, our results reveal how nutrient conditions impact the fitness of a major human pathogen in its environmental reservoir.

## Introduction

*Vibrio cholerae* is a Gram-negative, motile bacterium that primarily inhabits marine environments. When ingested, some serogroups are capable of causing the disease cholera, which is characterized by a profuse secretory diarrhea [1]. The differences between the marine environment and the human gut are profound, and *V. cholerae* must express distinct sets of genes in order to survive the challenges specific to each niche [2]. In the marine environment, this organism's expression profile is heavily influenced by contact with chitin, an insoluble polymer of β1,4-linked *N*-acetylglucosamine (GlcNAc) that is found in the shells of crustaceans [3]. *V. cholerae* frequently colonizes and forms biofilms on chitinous surfaces [3,4]. Chitin oligomers serve as both a nutrient source and an environmental cue that induces the transcription of genes required for efficient chitin catabolism as well as for behaviors that enhance its fitness in this communal setting [3,5]. These behaviors include chitin utilization itself as well as natural transformation, a mode of horizontal gene transfer in which bacteria take up exogenous DNA and integrate it into their genomes via homologous recombination [6]. Natural transformation can promote the spread of virulence factors and antibiotic resistance genes in bacterial pathogens [7]. The chitin response is therefore both clinically and ecologically significant for *V. cholerae*. The extracellular degradation of insoluble chitin and subsequent uptake of soluble chitin oligosaccharides is energetically costly; dozens of proteins are required for catabolism, and the enzymes secreted for external digestion may be exploited by cells other than the producer [8,9]. In contrast, simple six-carbon sugars (i.e., glucose) can enter glycolysis with minimal modification [10]. It may therefore be advantageous for bacteria to consume simple sugars before expending energy to degrade complex polysaccharides like chitin. This hierarchical consumption of sugars is typically achieved through upregulation of genes required to consume the preferred carbon source, repression of genes and proteins involved in catabolism of less preferred carbon sources, or both. This behavior is known as carbon catabolite repression (CCR) [11,12].

CCR is primarily mediated through proteins of the phosphoenolpyruvate-dependent phosphotransferase system (PTS). This system consists of a broadly conserved set of proteins that mediate a phosphorelay that ultimately facilitates the uptake and concomitant phosphorylation of specific carbohydrates [13]. The first protein in the PTS cascade is Enzyme I (EI). This protein autophosphorylates using the glycolytic intermediate phosphoenolpyruvate (PEP) as a substrate [12]. EI then transfers this phosphate to the histidine phosphocarrier protein (Hpr), which in turn transfers the phosphate to the EIIA domain of a sugar permease complex known

as Enzyme II (EII). The EIIA domain can then transfer the phosphate to a cognate EIIB domain, which then transfers the phosphate to the carbohydrate substrate during uptake through the EIIC(D) permease. During substrate import, phosphates flow rapidly through the PTS, skewing the intermediate proteins towards their dephosphorylated states. In the absence of substrate, phosphate flow halts, and the intermediates accumulate in their phosphorylated form [12,14]. Importantly, this phenomenon means that the phosphorylation status of PTS intermediates reflects the availability of their respective substrates [12,14]. The relationship between the phosphorylation status of PTS proteins and carbohydrate availability is readily exploited by the cell. PTS intermediates perform a host of regulatory functions that enable careful modulation of cellular processes in response to nutrient availability. These functions depend on the phosphorylation state of PTS intermediates. While many PTS proteins have documented regulatory roles, the central mediator of CCR in Gram-negative bacteria is the glucose-specific Enzyme IIA (EIIA$^{Glc}$), which in *V. cholerae* is responsible for the import of glucose as well as sucrose, trehalose, and *N*-acetylglucosamine [12,15]. Glucose-induced dephosphorylation of EIIA$^{Glc}$ enables its participation in an array of regulatory activities [12]. Regulation by EIIA$^{Glc}$ has been described for behaviors including metabolism [16], virulence [17], motility [18], and biofilm formation [19]. Understanding the different mechanisms underlying EIIA$^{Glc}$ activity is critical to unraveling how these crucial processes are influenced by carbon availability.

The *V. cholerae* chitin response is among the many behaviors regulated by CCR. *V. cholerae* senses and responds to chitin through the noncanonical hybrid sensor kinase ChiS (<u>chi</u>tin <u>sen</u>sor) [8]. In the presence of chitin, ChiS activates the transcription of genes required for chitin utilization and natural transformation [8,20,21]. However, it has been observed that *V. cholerae* cells grown on EIIA$^{Glc}$-dependent sugars cannot undergo chitin-induced natural transformation [22]. Similarly, strains expressing dephosphorylated EIIA$^{Glc}$ (dephospho-EIIA$^{Glc}$) are non-transformable [23]. The mechanism by which EIIA$^{Glc}$ inhibits this important cellular process is unknown, but a recent study has shown that dephospho-EIIA$^{Glc}$ can reduce ChiS-dependent transcriptional activation [23]. This finding raises the possibility that EIIA$^{Glc}$ inhibition of natural transformation may involve repression of ChiS itself. The mechanism by which dephospho-EIIA$^{Glc}$ represses ChiS is unknown, and it is not clear whether such repression contributes to the previously described CCR of natural transformation [22,23]. As ChiS also activates the genes required for chitin utilization, it is also possible that dephospho-EIIA$^{Glc}$ will prevent efficient growth on chitin, which is critical for *Vibrio* survival in marine environments [3]. Such regulation might enable *V. cholerae* cells in a chitin-associated biofilm to rapidly take advantage of transiently available sugars (perhaps released from decomposing wildlife).

In this study, we identify the molecular mechanism of CCR of natural transformation and chitin utilization in *V. cholerae*. We find that dephospho-EIIA$^{Glc}$ interacts with ChiS, and that this interaction is required for repression to occur. ChiS suppressor mutations that escape dephospho-EIIA$^{Glc}$-dependent repression also relieve CCR of chitin utilization and natural transformation. Together, these findings suggest that repression of ChiS by dephospho-EIIA$^{Glc}$ is the dominant mechanism of CCR for chitin-dependent responses in *V. cholerae*.

## Results

### Dephosphorylated EIIA$^{Glc}$ represses ChiS transcriptional activity and downstream behaviors

ChiS activity can be assessed through its direct activation of the chitobiose utilization operon (*chb*) [8,24]. In the presence of chitin, ChiS strongly activates this locus, resulting in the

transcription of genes required for the uptake and utilization of chitobiose [8]. In the absence of chitin, ChiS is repressed by the periplasmic chitin binding protein (CBP). Previous work has shown that the deletion of CBP is sufficient to genetically activate ChiS even in the absence of chitin [8]. We therefore used a P$_{chb}$-*GFP* transcriptional reporter [25] in a Δ*cbp* background to interrogate ChiS activity. Consistent with previous findings [23], we observed that rendering EIIA$^{Glc}$ dephosphorylated by the deletion of its upstream phosphate donor EI represses ChiS transcriptional activity (**Fig 1A**). This repression can be relieved by the simultaneous deletion of EIIA$^{Glc}$, suggesting that EIIA$^{Glc}$ plays a critical role in this repression (**Fig 1A**). We generated a genetically locked dephospho-EIIA$^{Glc}$ allele [26] by mutating the phosphorylated histidine to a glutamate (H91Q) and found that this allele was sufficient to repress ChiS activity in a background where EI was intact (**Fig 1A**). Furthermore, a proposed phosphomimetic EIIA$^{Glc}$ H91D allele [19] cannot repress ChiS in a ΔEI background. All together, these results are consistent with a model in which dephospho-EIIA$^{Glc}$ represses ChiS activity (**Fig 1A**).

These findings suggested that the phosphorylation status of EIIA$^{Glc}$ is central to its ability to repress ChiS activity. Phosphorylated EIIA$^{Glc}$ stimulates the activity of adenylate cyclase to increase the cellular levels of the second messenger cyclic AMP (cAMP), while dephospho-EIIA$^{Glc}$ cannot promote this activity [12,27–29]. ChiS-dependent activity requires cAMP, as evidenced by the fact that an adenylate cyclase mutant displays no induction of P$_{chb}$ unless exogenous cAMP is supplied (**S1A Fig**). Because the deletion of EIIA$^{Glc}$ markedly rescues ChiS activity, we reasoned that low cAMP levels were not the primary mechanism for ΔEI repression of ChiS since ΔEI ΔEIIA$^{Glc}$ mutants and ΔEI mutants both lack phosphorylated EIIA$^{Glc}$ for stimulating adenylate cyclase activity. However, to formally test this, we performed assays in the presence and absence of supplemental cAMP. Consistent with previous reports [23], cAMP is unable to rescue ChiS activity in the ΔEI or EIIA$^{Glc}$ H91Q backgrounds (**S1A Fig**). Thus, EIIA$^{Glc}$ is repressing ChiS activity via a mechanism other than through the indirect regulation of adenylate cyclase activity because cAMP cannot rescue ChiS activity while dephospho-EIIA$^{Glc}$ is present in the cell.

Our previous experiments utilized a Δ*cbp* mutation to genetically induce ChiS activity in the absence of chitin. Therefore, we next asked whether repression by EIIA$^{Glc}$ affects ChiS transcriptional activity in a more physiologically relevant setting. To test this question, we grew cells containing a P$_{chb}$-*mCherry* reporter [25] on insoluble chitin for 48 hours and assessed fluorescence of individual cells using microscopy. The mCherry signal was normalized to a constitutively expressed GFP construct to account for internal noise in gene expression. Importantly, these strains all had *cbp* intact, which allows for natural chitin-induced activation of ChiS. As expected, parent cells robustly activated the P$_{chb}$-*mCherry* reporter under these conditions, while cells lacking ChiS did not (**Fig 1B**). The ΔEI mutation significantly repressed ChiS activity in this assay (**Fig 1B**). This repression could be fully relieved by the tandem deletion of EIIA$^{Glc}$ (**Fig 1B**), mirroring what was observed when the Δ*cbp* mutation was used to induce ChiS activity (**Fig 1A**). Interestingly, the nonphosphorylatable EIIA$^{Glc}$ H91Q allele only slightly repressed ChiS in this assay (**Fig 1B**). This result suggests that EIIA$^{Glc}$ H91Q may be a less potent repressor of ChiS than natural dephospho-EIIA$^{Glc}$, which is more readily observed when ChiS activity is stimulated via natural chitin induction. The reduced efficacy of the H91Q allele might be explained by the fact that for other targets of EIIA$^{Glc}$, the H91 residue is part of the protein-protein binding interface [30–33]. Thus, beyond affecting phosphorylation, mutations to this residue may affect the ability of EIIA$^{Glc}$ to interact with its downstream targets, which may include ChiS. Lastly, the phosphomimetic EIIA$^{Glc}$ H91D [19] allele mediated no repression of ChiS in the ΔEI background (**Fig 1B**), further supporting that dephospho-EIIA$^{Glc}$ inhibits ChiS activity. Once again, exogenous addition of cAMP was unable to rescue ChiS activity in the ΔEI background (**Figs 1B and S1B**). Interestingly, the

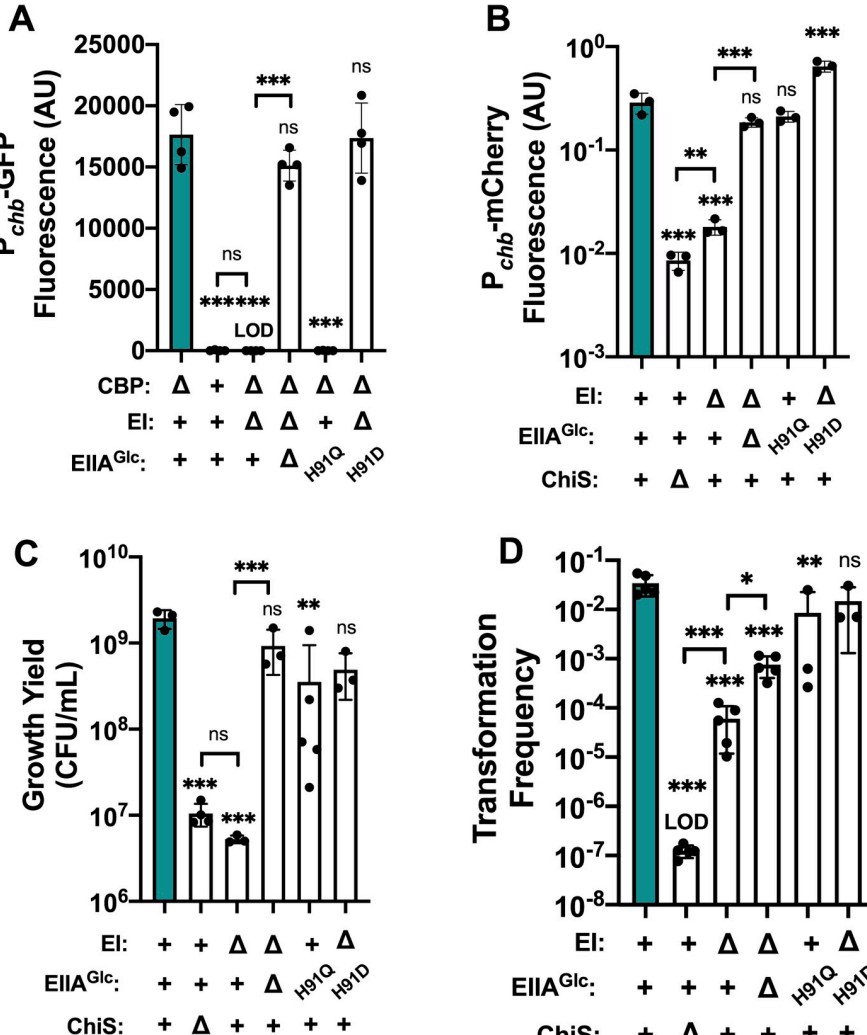

**Fig 1. Dephosphorylated EIIA<sup>Glc</sup> represses ChiS transcriptional activity and downstream behaviors. (A)** ChiS activation of the *chb* promoter ($P_{chb}$) was assessed in rich medium (in the absence of chitin) via a $P_{chb}$-GFP reporter in the indicated strain backgrounds. **(B)** ChiS activation of $P_{chb}$-mCherry was assessed in the indicated strains after cells were incubated on chitin for 48 hrs. mCherry signal was normalized to a constitutively expressed GFP construct. **(C)** Final growth yield of the indicated strains after a 72 hr incubation in M9 minimal medium containing chitin as the sole carbon source. **(D)** Chitin-induced natural transformation assays of the indicated strains. In all experiments, reactions were supplemented with 5 mM cAMP. Results are from at least three independent biological replicates and shown as the mean ± SD. Statistical comparisons were made by one-way ANOVA with Tukey's multiple comparison test. NS, not significant. *** = $p < 0.001$, ** = $p < 0.01$, * = $p < 0.05$. LOD, limit of detection. Statistical identifiers directly above bars represent comparisons to the parent (teal, first bar).

addition of cAMP did restore $P_{chb}$ expression in the ΔEI/EIIA<sup>Glc</sup> strain back to parent levels of activity (**S1B Fig**), suggesting that a reduction of cAMP in this background (due to low adenylate cyclase activity in the absence of phosphorylated EIIA<sup>Glc</sup>) does play a small role in limiting ChiS transcriptional activity. Together, these results support our previous observations and reaffirm that EIIA<sup>Glc</sup> inhibits ChiS activity in physiologically relevant conditions.

Having observed a strong repressive effect on ChiS-dependent transcription, we sought to determine whether EIIA<sup>Glc</sup> repression impacts ChiS-controlled behaviors. ChiS directly activates the genes required for chitin uptake and catabolism [3,8,24]. We therefore asked whether

dephospho-EIIA$^{Glc}$ affects the ability of *V. cholerae* to grow on insoluble chitin over 72 hr. We found that deletion of EI reduced final growth yield to the same level as a ChiS deletion (**Fig 1C**). This defect in growth yield was restored in the ΔEI ΔEIIA$^{Glc}$ double mutant (**Fig 1C**). Growth yield was only partially reduced by EIIA$^{Glc}$ H91Q, which is consistent with the intermediate repression mediated by this allele on chitin-induced ChiS activity. EIIA$^{Glc}$ H91D relieved the repressive effects of the ΔEI mutation on growth yield (**Fig 1C**). Once again, exogenous cAMP did not alter these outcomes (**S2 Fig**). These results suggest that EIIA$^{Glc}$-mediated repression of ChiS activity is sufficient to repress the ability of *V. cholerae* to use insoluble chitin as a carbon source.

Apart from regulating chitin utilization, ChiS is also essential for the induction of natural transformation [5,20,21], a physiological state in which cells are able to take up extracellular DNA from the environment to incorporate into their genome by homologous recombination [6]. Consistent with previous work, we found that while parent cells transform at a high frequency, transformation is significantly impaired in a ΔEI background [23] (**Fig 1D**). However, in contrast with previous work [23], we observed a partial rescue in transformation in the ΔEI ΔEIIA$^{Glc}$ double mutant. We attribute this discrepancy to differences in the dynamic ranges of our transformation assays; the reduced transformation frequency of the ΔEI ΔEIIA$^{Glc}$ strain likely falls below the limit of detection of the less sensitive assay used to previously test this question. Interestingly, the EIIA$^{Glc}$ H91Q mutant displayed a subtle but statistically significant transformation defect (**Fig 1D**). Our findings above indicate that this allele is likely a less effective repressor of ChiS activity compared to natural dephospho-EIIA$^{Glc}$. Previous work from our lab has also shown that strains that exhibit growth defects on chitin can still transform at high levels, suggesting that the threshold of chitin induction required for cells to transform may be lower than that required for growth on chitin [34]. This may explain why the EIIA$^{Glc}$ H91Q mutant exhibited a subtle transformation defect (**Fig 1D**). The phosphomimetic EIIA$^{Glc}$ H91D mutation in the ΔEI background transformed at parent levels, which further suggests that dephospho-EIIAGlc is required for inhibition of natural transformation in the ΔEI background (**Fig 1D**). As seen in the previous assays, cAMP was unable to restore transformation of the ΔEI mutant to parent levels (**S3 Fig**). All together, these results indicate that dephospho-EIIA$^{Glc}$ significantly impairs ChiS transcriptional activity and downstream behaviors (*i.e.*, growth on chitin and chitin-induced natural transformation), in a cAMP-independent manner.

## Dephosphorylated EIIA$^{Glc}$ inhibits ChiS DNA-binding activity

We next wanted to probe the mechanism by which EIIA$^{Glc}$ represses ChiS activity. ChiS has a DNA-binding domain and directly binds to the *chb* promoter [24,35]. Thus, we hypothesized that EIIA$^{Glc}$ might interfere with ChiS DNA-binding. To test this possibility, we performed chromatin immunoprecipitation (ChIP)-qPCR assays using a previously characterized functional [24] FLAG-tagged construct of ChiS to assess DNA-binding *in vivo*. We hypothesized that if dephospho-EIIA$^{Glc}$ inhibits DNA binding, ChiS should no longer be able to ChIP the *chb* promoter; however, if dephospho-EIIA$^{Glc}$ inhibits activation downstream of DNA binding, we should still observe ChiS DNA-binding in a ΔEI mutant. When we performed this experiment, we observed that ChiS DNA binding is inhibited in a ΔEI mutant but rescued in the ΔEI ΔEIIA$^{Glc}$ mutant. These data suggest that dephospho-EIIA$^{Glc}$ inhibits the DNA-binding activity of ChiS (**Fig 2**).

## ChiS interacts with dephosphorylated EIIA$^{Glc}$

We next sought to determine the mechanism by which dephospho-EIIA$^{Glc}$ represses ChiS DNA-binding activity. EIIA$^{Glc}$ has been shown to regulate many target proteins through

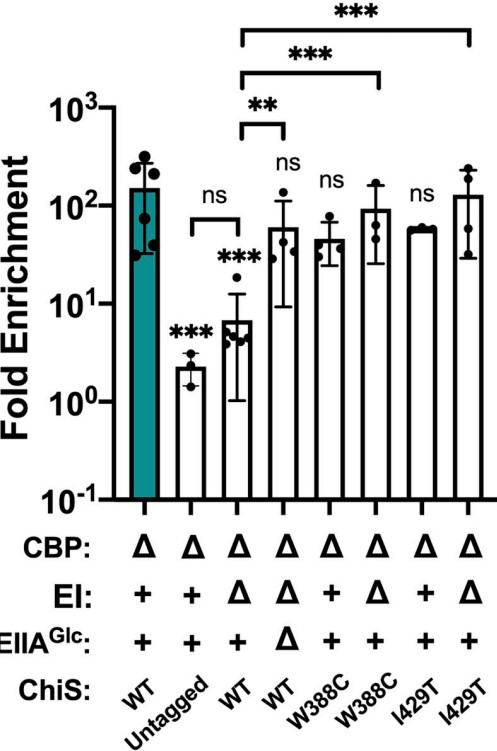

**Fig 2. Dephosphorylated EIIA^Glc inhibits the DNA-binding activity of ChiS^WT but not ChiS^W388C or ChiS^I429T in vivo.** ChIP-qPCR assays were performed on the indicated strains to assess ChiS binding to the *chb* promoter (P_{chb}) *in vivo*. Fold enrichment of P_{chb} relative to rpoB is shown. Results are from three independent biological replicates and shown as the mean ± SD. Statistical comparisons were made by one-way ANOVA with Tukey's multiple comparison test. NS, not significant. *** = $p < 0.001$, ** = $p < 0.01$, * = $p < 0.05$. Statistical identifiers directly above bars represent comparisons to the parent (teal, first bar).

protein-protein interactions, in a manner that can either stimulate or inhibit the activity of those interacting partners [16,30,36,37]. We therefore hypothesized that dephospho-EIIA^Glc might interact with ChiS to repress its activity. To test whether EIIA^Glc and ChiS interact, we first assessed whether these proteins colocalized *in vivo*. Specifically, we generated cells that express an EIIA^Glc-mCherry fusion that can still repress ChiS (**S4 Fig**) and an inducible GFP-tagged construct of ChiS [24], and examined their cellular localization pattern using fluorescence microscopy. We observed that when we did not induce ChiS-GFP, EIIA^Glc-mCherry was diffuse throughout the cytoplasm (**Fig 3**). When we overexpressed ChiS in cells where EI is intact, EIIA^Glc remained diffuse in 90% of cells, while in 10% of cells, ChiS and EIIA^Glc formed colocalized foci (**Fig 3**). Strikingly, when we deleted EI (rendering all EIIA^Glc within the cell dephosphorylated), the percentage of cells containing colocalized ChiS-EIIA^Glc foci increased to 90% (**Fig 3**). Furthermore, 81% of cells expressing EIIA^Glc H91Q displayed colocalizing foci (**Fig 3**), further suggesting that dephospho-EIIA^Glc interacts with ChiS. Consistent with this hypothesis, only 4% of cells expressing the phosphomimetic EIIA^Glc H91D allele in a ΔEI background display colocalizing foci.

Colocalization of proteins *in vivo* only indicates that they are occupying a similar subcellular site and cannot independently confirm that they interact. We therefore wanted to test whether the colocalization observed between ChiS and EIIA^Glc was the result of an interaction. To this end, we took advantage of the PopZ-linked apical recruitment (POLAR) assay [38]. In

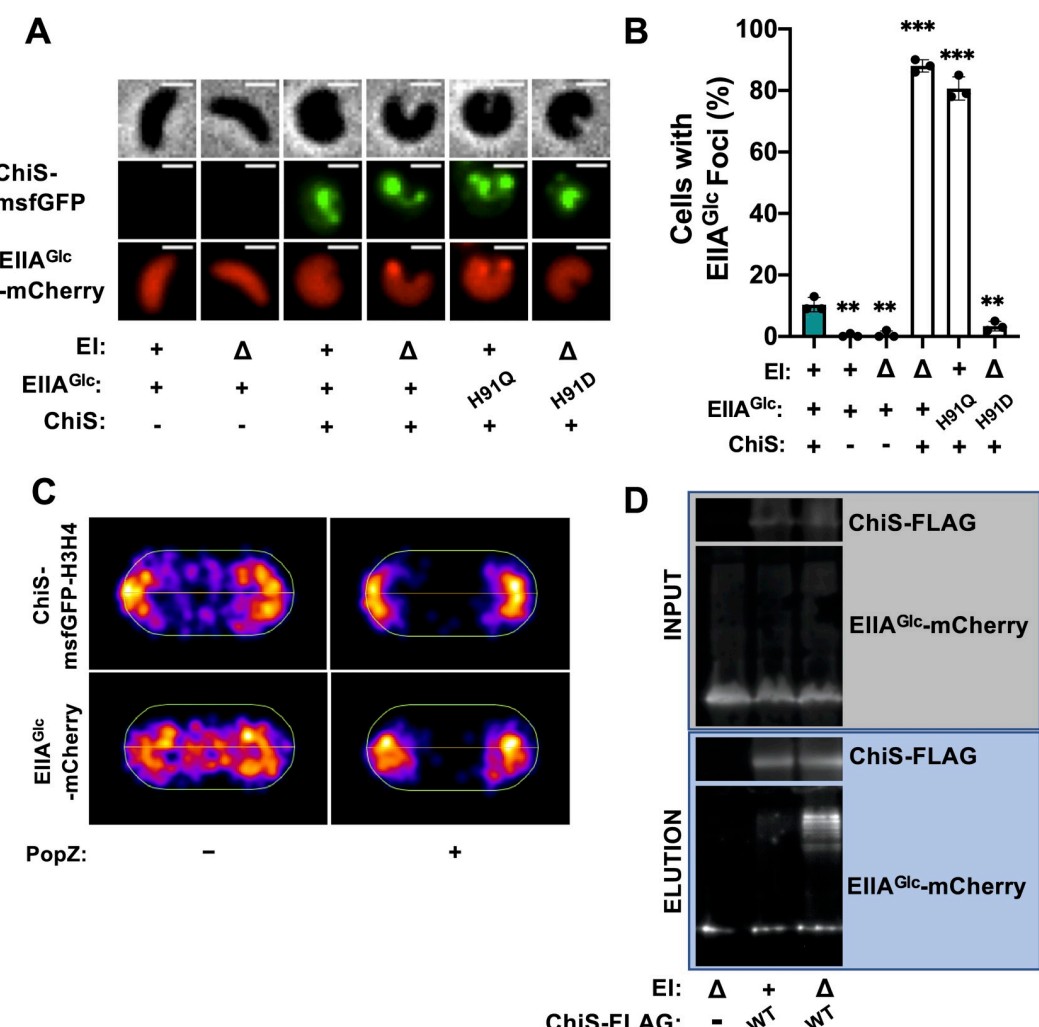

**Fig 3. ChiS interacts with dephosphorylated EIIA^Glc.** Colocalization of ChiS and EIIA$^{Glc}$ was assessed in cells either containing or lacking a P$_{tac}$-*chiS-msfGFP* construct and a functional EIIA$^{Glc}$-*mCherry* fusion at the native locus. For strains containing a P$_{tac}$-*chiS-msfGFP* construct, cultures were supplemented with 20 μM IPTG. (**A**) Representative images of ChiS-msfGFP and EIIA$^{Glc}$-mCherry localization in the indicated strain backgrounds. Phase (top) is shown to demarcate cell boundaries, GFP fluorescence (middle) shows ChiS-msfGFP localization, and mCherry fluorescence (bottom) shows EIIA$^{Glc}$-mCherry localization. Scale bar, 1 μm. (**B**) Quantification of the percentage of cells containing EIIA$^{Glc}$ foci in the indicated strains ($n$ = 300 cells analyzed per strain). All EIIA$^{Glc}$ foci observed colocalized with ChiS as shown in **A**. Data are from three independent biological replicates and shown as the mean ± SD. Statistical comparisons were made by one-way ANOVA with Tukey's multiple comparison test. NS, not significant. *** = $p < 0.001$, ** = $p < 0.01$, * = $p < 0.05$. Statistical identifiers directly above bars represent comparisons to the parent (teal, first bar). (**C**) Representative heat maps showing the localization of EIIA$^{Glc}$-mCherry and ChiS-H3H4-msfGFP foci in ΔEI cells in the presence and absence of PopZ expression. $n \geq 300$ cells analyzed per condition. Data are representative of two independent experiments. (**D**) Representative image of pulldown assays. DSS crosslinked lysates from cells expressing EIIA$^{Glc}$-mCherry and ChiS-FLAG were immunoprecipitated with anti-FLAG magnetic beads. Bound protein was eluted and protein input (grey box) and elution (blue box) samples were visualized by western blotting. Data are representative of two independent experiments.

this assay, two putative interaction partners–the "bait" and "prey"—are fluorescently tagged. The bait is also tagged with an H3H4 PopZ oligomerization domain. The H3H4 domain specifically interacts with the PopZ protein from *Caulobacter crescentus*, which localizes to cell poles when heterologously expressed in other bacterial species [39]. Thus, when PopZ is expressed in cells it relocalizes the bait to the cell poles. If the bait and prey interact, PopZ-dependent relocalization of the bait should cause a concomitant relocalization of the prey to

the cell poles. However, if the bait and prey do not interact, PopZ-dependent relocalization of the bait should not alter the localization of the prey. Thus, to test if ChiS and EIIA$^{Glc}$ interact, we generated cells with ChiS-H3H4-msfGFP as the bait and EIIA$^{Glc}$-mCherry as the prey. In the absence of PopZ, ChiS and EIIA$^{Glc}$ foci localize both to the cell poles and midcell (**Fig 3C**). When PopZ is induced, the ChiS bait relocalizes to the cell poles as expected (**Fig 3C**). Consistent with ChiS and EIIA$^{Glc}$ interacting, nearly all EIIA$^{Glc}$ foci are relocalized to the cell poles when PopZ is induced (**Fig 3C**). Together, these results strongly suggest that ChiS and dephospho-EIIA$^{Glc}$ interact.

To further test this interaction, we also performed *in vivo* pulldowns using the functional ChiS-FLAG and EIIA$^{Glc}$-mCherry alleles. To stabilize protein-protein interactions, the crosslinker disuccinimidyl suberate (DSS) was added to cells prior to performing the pull down. To test whether ChiS interacts with EIIA$^{Glc}$ in a phosphorylation-dependent manner, we performed pulldowns of ChiS-FLAG in strains where EI was intact vs deleted. We observed that when EI was intact, no EIIA$^{Glc}$-mCherry could be detected in the elution fraction (**Fig 3D**). However, when EI was deleted, a high molecular weight band was detected in the EIIA$^{Glc}$-mCherry elution fraction, consistent with an interaction between EIIA$^{Glc}$ and ChiS (**Fig 3D**). This band was not detected in the absence of ChiS, even if EI was deleted (**Fig 3D**). Importantly, we confirmed that dephospho-EIIA$^{Glc}$ represses ChiS activity in the same strains and conditions used for these pulldown assays using the P$_{chb}$-*GFP* reporter (**S5 Fig**). These results further indicate that ChiS and dephospho-EIIA$^{Glc}$ interact.

## Isolation of ChiS suppressor alleles that cannot be inhibited by dephosphorylated EIIA$^{Glc}$

The genotypes that promote ChiS-EIIA$^{Glc}$ interactions (e.g., ΔEI and EIIA$^{Glc}$ H91Q) correlate with the genotypes in which EIIA$^{Glc}$ represses ChiS activity (**Figs 1A–1D and 3**). We therefore hypothesized that this interaction is required for EIIA$^{Glc}$ to inhibit ChiS. To test this possibility, we took an unbiased approach by isolating ChiS suppressor mutants that can escape EIIA$^{Glc}$-dependent inhibition and then tested these alleles for their ability to interact with EIIA$^{Glc}$. To isolate ChiS suppressor mutants, we used error-prone PCR to randomly mutagenize ChiS and introduced these mutated alleles into a Δ*cbp* ΔEI (*i.e.*, dephospho-EIIA$^{Glc}$) strain containing a previously described P$_{chb}$-*lacZ* reporter [25,40]. We screened ~5,500 colonies for the recovery of ChiS activity as indicated by blue colonies on X-gal plates. Of the suppressor mutations we isolated, two were able to fully rescue ChiS transcriptional activity in a ΔEI background: W388C and I429T (**Fig 4**). Both suppressor alleles significantly rescued transcriptional activity using either Δ*cbp* or insoluble chitin to induce ChiS activity in the ΔEI background (**Fig 4A and 4B**). This rescue was similar to that conferred by the deletion of EIIA$^{Glc}$ (ΔEI ΔEIIA$^{Glc}$) (**Fig 4A and 4B**). Furthermore, deleting EIIA$^{Glc}$ did not further increase the activity of these suppressor alleles in the ΔEI background, implying that the ChiS suppressor mutations and EIIA$^{Glc}$ are epistatic and work via the same pathway (**Fig 4A and 4B**). Consistent with transcriptional activity, dephospho-EIIA$^{Glc}$ failed to inhibit the DNA-binding activity of both ChiS suppressor alleles *in vivo* (**Fig 2**). Importantly, the expression level of both ChiS suppressor mutants is comparable to the WT in a ΔEI background, suggesting that an increased abundance of ChiS cannot account for the phenotypes observed (**S6 Fig**).

We also tested the ability of CBP to repress both ChiS suppressor alleles. Like native ChiS, ChiS$^{W388C}$ was fully repressed by CBP (**Fig 4A**). ChiS$^{I429T}$ had observable activity in the presence of CBP, but its activity was still significantly increased when CBP was deleted (**Fig 4A**). This finding suggests that ChiS$^{I429T}$ can still be repressed by CBP, but not as efficiently as

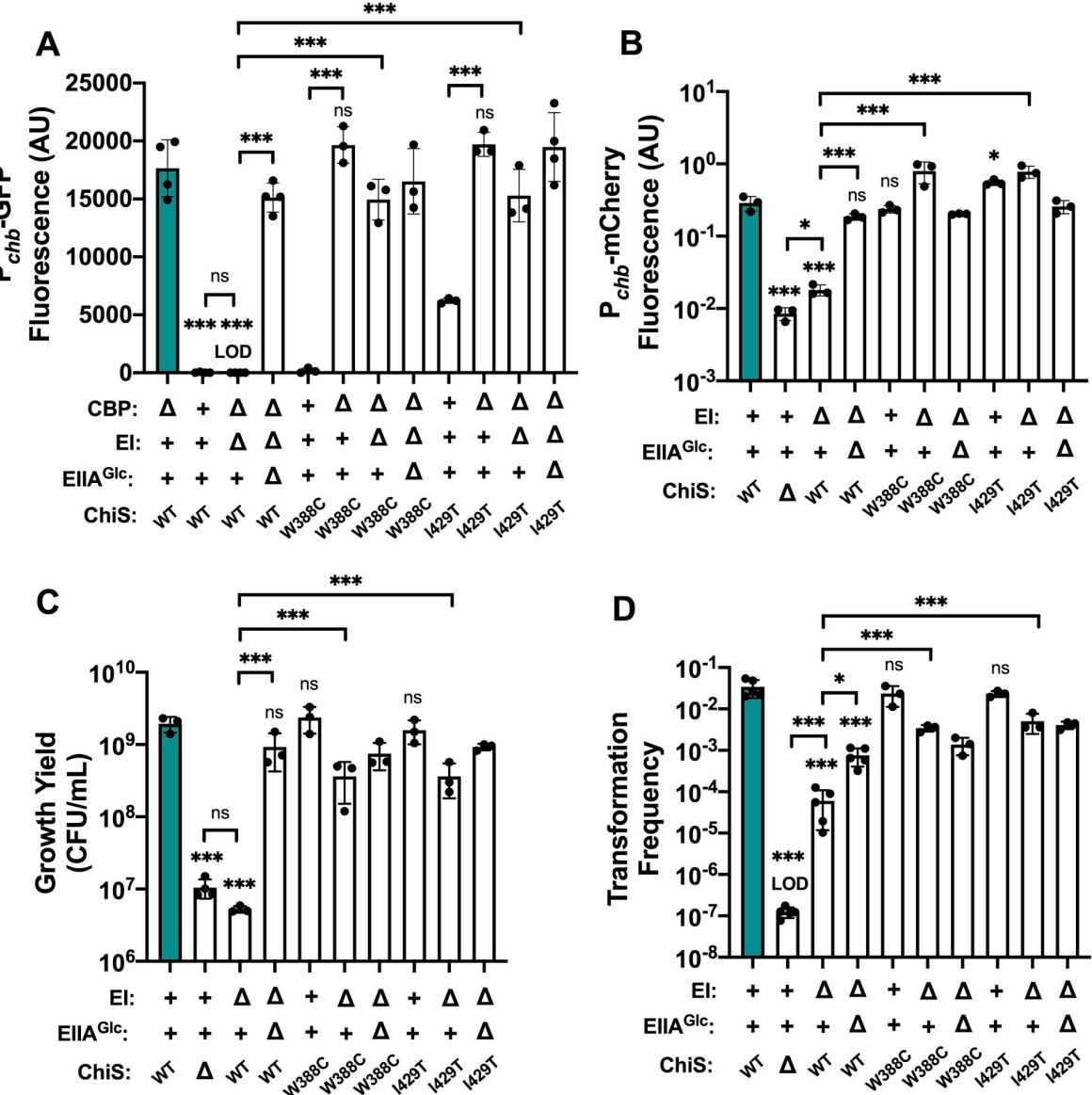

**Fig 4. Dephosphorylated EIIA<sup>Glc</sup> cannot inhibit the activity of ChiS suppressor alleles.** (**A**) ChiS activation of the *chb* promoter (P$_{chb}$) was assessed in rich medium (in the absence of chitin) using a P$_{chb}$-GFP reporter in the indicated strain backgrounds. (**B**) ChiS activation of P$_{chb}$-mCherry was assessed in the indicated strains after cells were incubated on chitin for 48 hrs. mCherry signal was normalized to a constitutively expressed GFP construct. (**C**) Final growth yield of the indicated strains after a 72 hr incubation in M9 minimal media containing chitin as the sole carbon source. (**D**) Chitin-induced natural transformation assays of the indicated strains. In all experiments, reactions were supplemented with 5 mM cAMP. Results are from at least three independent biological replicates and shown as the mean ± SD. Statistical comparisons were made by one-way ANOVA with Tukey's multiple comparison test. NS, not significant. *** = $p < 0.001$, ** = $p < 0.01$, * = $p < 0.05$. LOD, limit of detection. Statistical identifiers directly above bars represent comparisons to the parent (teal, first bar). The first four bars in each panel are the same data presented in **Fig 1** and are included here for ease of comparison.

native ChiS. Together, these data suggest that the W388C and I429T mutations render ChiS specifically resistant to repression by dephospho-EIIA<sup>Glc</sup>.

Because these suppressor alleles restored ChiS transcriptional activity in the ΔEI mutant, we next tested whether they also rescue the ChiS-dependent phenotypes of chitin utilization and natural transformation from ΔEI repression. We found that both suppressor mutations

significantly rescued both chitin utilization (**Fig 4C**) and natural transformation (**Fig 4D**). These data suggest that the repression of these behaviors by the ΔEI mutation is largely mediated through EIIA$^{Glc}$-dependent repression of ChiS activity. Furthermore, these outcomes were largely unaffected by the addition of cAMP, suggesting that dephospho-EIIA$^{Glc}$-dependent repression of ChiS activity is the primary mechanism of CCR in the ΔEI background (**S2 and S3 Figs**).

## ChiS suppressor alleles no longer interact with dephosphorylated EIIA$^{Glc}$

We previously showed that dephospho-EIIA$^{Glc}$ and ChiS colocalize *in vivo* (**Fig 3**). We hypothesized that if EIIA$^{Glc}$ needs to interact with ChiS to inhibit its activity, then the ChiS suppressor mutants would no longer be able to interact with dephospho-EIIA$^{Glc}$. Indeed, we found that these ChiS suppressor alleles (W388C and I429T) significantly reduced the colocalization of ChiS with dephospho-EIIA$^{Glc}$ when compared to the ChiS parent (**Fig 5A and 5B**). Furthermore, dephospho-EIIA$^{Glc}$-mCherry no longer co-immunoprecipitated with ChiS$^{W388C}$-FLAG (**Fig 5C**). These data strongly indicate that EIIA$^{Glc}$ must interact with ChiS to repress its activity.

## The ChiS$^{W388C}$ suppressor allele confers resistance to glucose-induced CCR of natural transformation

Thus far, we have induced CCR via genetics means (i.e., via use of ΔEI or EIIA$^{Glc}$ H91Q mutations) to show that dephospho-EIIA$^{Glc}$ inhibits ChiS activity. We next sought to test CCR of ChiS using carbon sources that naturally induce CCR. It was previously shown that glucose inhibits natural transformation in *V. cholerae* [5,22]. Because cAMP/CRP are critical for induction of natural transformation in conjunction with the master competence regulator TfoX [41,42], this inhibition was previously assumed to be due to the reduction of cAMP resulting from the dephosphorylation of EIIA$^{Glc}$ by glucose [41–44]. We observed that while cAMP does increase transformation in a ΔEI strain, transformation is increased further by the deletion of EIIA$^{Glc}$ (**S3 Fig**). Furthermore, deletion of EIIA$^{Glc}$ is sufficient to increase transformation even in the absence of exogenous cAMP (**S3 Fig**). Therefore, we hypothesized that the dominant mechanism for CCR of natural transformation by glucose is through the repression of ChiS activity by dephospho-EIIA$^{Glc}$. To test this, we compared the transformation efficiency of a strain with ChiS$^{WT}$ to a strain with ChiS$^{W388C}$ in the presence of glucose to induce CCR. As previously reported, glucose inhibited transformation of our parent strain (i.e., harboring ChiS$^{WT}$) [5,22] (**S7A Fig**). However, contrary to our initial hypothesis, we found that transformation of ChiS$^{W388C}$ was also inhibited by glucose (**S7A Fig**). Furthermore, the addition of exogenous cAMP did not rescue transformation in either strain background when glucose was present (**S7A Fig**). This is in contrast to the rescue conferred by exogenous cAMP on the transformation frequency of a mutant lacking adenylate cyclase (**S3 Fig**).

One possible explanation for these results is that catabolism and growth of cells on glucose during these transformation assays inhibits transformation nonspecifically—for example, through dilution of the machinery required for DNA uptake and integration during rapid cell division. If this hypothesis is true, other supplemental carbon sources should be similarly repressive—even if they are not imported by the PTS (and therefore, would not activate a canonical CCR response). To test this, we assessed transformation efficiency in the presence of fructose (a PTS sugar that uses a separate EIIA protein), maltose (an ABC transported sugar), and a mixture of glutamate and proline. We found that all carbon sources significantly inhibited the transformation of both ChiS$^{WT}$ and ChiS$^{W388C}$ regardless of whether exogenous cAMP was provided (**S7A Fig**). These data suggest that the growth of cells in the presence of

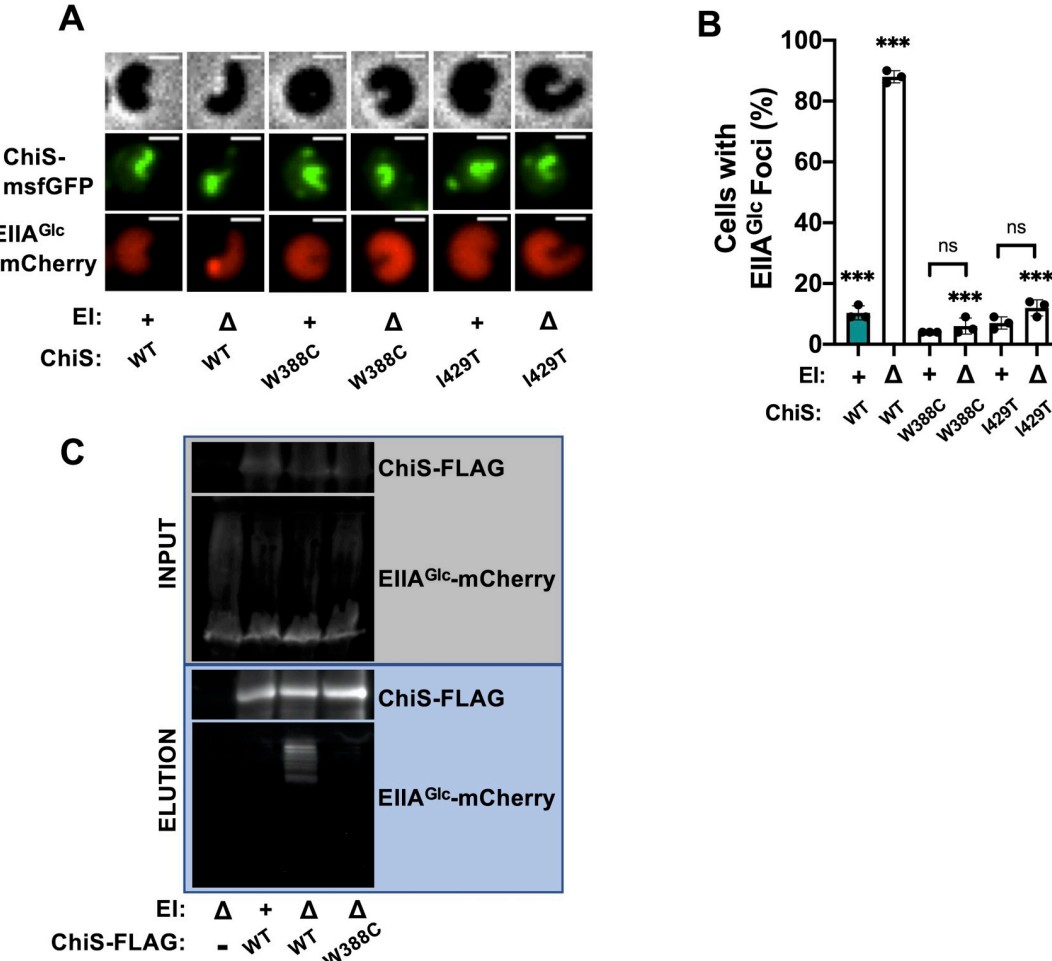

**Fig 5. ChiS suppressor alleles no longer interact with dephosphorylated EIIA$^{Glc}$.** Colocalization of ChiS and EIIA$^{Glc}$ was assessed in cells containing a P$_{tac}$-*chiS-msfGFP* construct and a functional EIIA$^{Glc}$-*mCherry* fusion at the native locus. Cultures were supplemented with 20 μM IPTG to induce expression of the indicated ChiS-msfGFP allele. (**A**) Representative images of cells to assess the localization of ChiS-msfGFP and EIIA$^{Glc}$-mCherry in the indicated strains. Phase (top) is shown to demarcate cell boundaries, GFP fluorescence (middle) shows ChiS-msfGFP localization, and mCherry fluorescence (bottom) shows EIIA$^{Glc}$-mCherry localization. Scale bar, 1 μm. (**B**) Quantification of the percent of cells containing EIIA$^{Glc}$ foci in the indicated strains ($n$ = 300 cells analyzed per strain). All EIIA$^{Glc}$ foci observed colocalized with ChiS as shown in **A**. Data are from three independent biological replicates and shown as the mean ± SD. Statistical comparisons were made by one-way ANOVA with Tukey's multiple comparison test. NS, not significant. *** = $p < 0.001$, ** = $p < 0.01$, * = $p < 0.05$. Statistical identifiers directly above bars represent comparisons to the parent (teal, first bar). Data for ChiS$^{WT}$ in **B** are the same data shown in **Fig 2B** and are included here for ease of comparison. (**C**) Representative image of pulldown assays. DSS crosslinked lysates from cells expressing EIIA$^{Glc}$-mCherry and the indicated allele of ChiS-FLAG were immunoprecipitated with anti-FLAG magnetic beads. Bound protein was eluted and protein input (grey box) and elution (blue box) samples were visualized by western blotting. Data are representative of two independent experiments.

an additional carbon source (whether it can induce CCR or not) confounds our ability to assess the role of CCR in repressing ChiS using natural substrates. We therefore devised a strategy to separate the effects of glucose induction of CCR from its downstream metabolism. Phosphoglucoisomerase (PGI) is responsible for converting glucose-6-phosphate into fructose-6-phosphate during glycolysis. Thus, ΔPGI mutants can still take up glucose via the PTS but lack the ability to catabolize it (**S8 Fig**). When we tested the transformation of a ΔPGI mutant, we found that it maintained the ability to transform. However, this transformation was fully inhibited by glucose in cells with ChiS$^{WT}$. By contrast, ΔPGI cells with ChiS$^{W388C}$

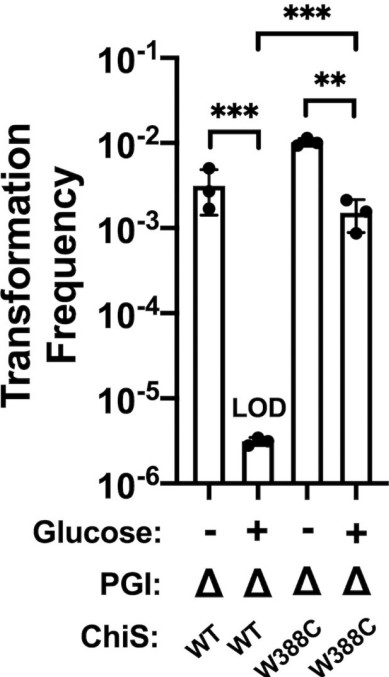

**Fig 6. The ChiS$^{W388C}$ allele confers resistance to glucose-induced CCR of natural transformation.** Chitin-induced natural transformation assays of the indicated strains. Transformation reactions were supplemented with 0.5% glucose as indicated. All transformation reactions were supplemented with 5 mM cAMP. Results are from at least three independent biological replicates and shown as the mean ± SD. Statistical comparisons were made by one-way ANOVA with Tukey's multiple comparison test. NS, not significant. *** = $p < 0.001$, ** = $p < 0.01$, * = $p < 0.05$. LOD, limit of detection.

were highly transformable (**Fig 6**). Once again, these phenotypes were not affected by cAMP (**S7B Fig**). These data suggest that glucose uptake causes CCR of natural transformation, and that the dominant mechanism for this repression is via EIIA$^{Glc}$-dependent inhibition of ChiS activity.

We next asked whether CCR of ChiS by EIIA$^{Glc}$ was specific to the E7946 strain of *V. cholerae* used throughout this study or if it was a conserved response. Specifically, we tested the ability of dephospho-EIIA$^{Glc}$ to repress ChiS activity in our chitin-induced reporter assay in two other toxigenic *V. cholerae* strains (C6706 [45], A1552 [46]) as well as a nontoxigenic, non-O1/O139 environmental isolate strain (CR03-424 [47]). Furthermore, we tested this response in the more distantly related species *Vibrio campbellii* DS40M4 [48,49]. In all strains tested, we observed ChiS-dependent activation of P$_{chb}$-*mCherry* in response to chitin (**S9 Fig**). Furthermore, we observed a significant decrease in activation upon the deletion of EI, which could be rescued by the simultaneous deletion of EIIA$^{Glc}$ (**S9 Fig**). These results demonstrate that dephospho-EIIA$^{Glc}$-dependent CCR of ChiS is a conserved response in these systems.

## Discussion

In this study, we uncover the molecular mechanism responsible for CCR of chitin-dependent responses in *V. cholerae* (**Fig 7**). We show that dephospho-EIIA$^{Glc}$ interacts with ChiS *in vivo* to repress its DNA-binding activity, resulting in the inhibition of chitin utilization and natural transformation. These results reveal a novel mechanism by which carbon availability regulates

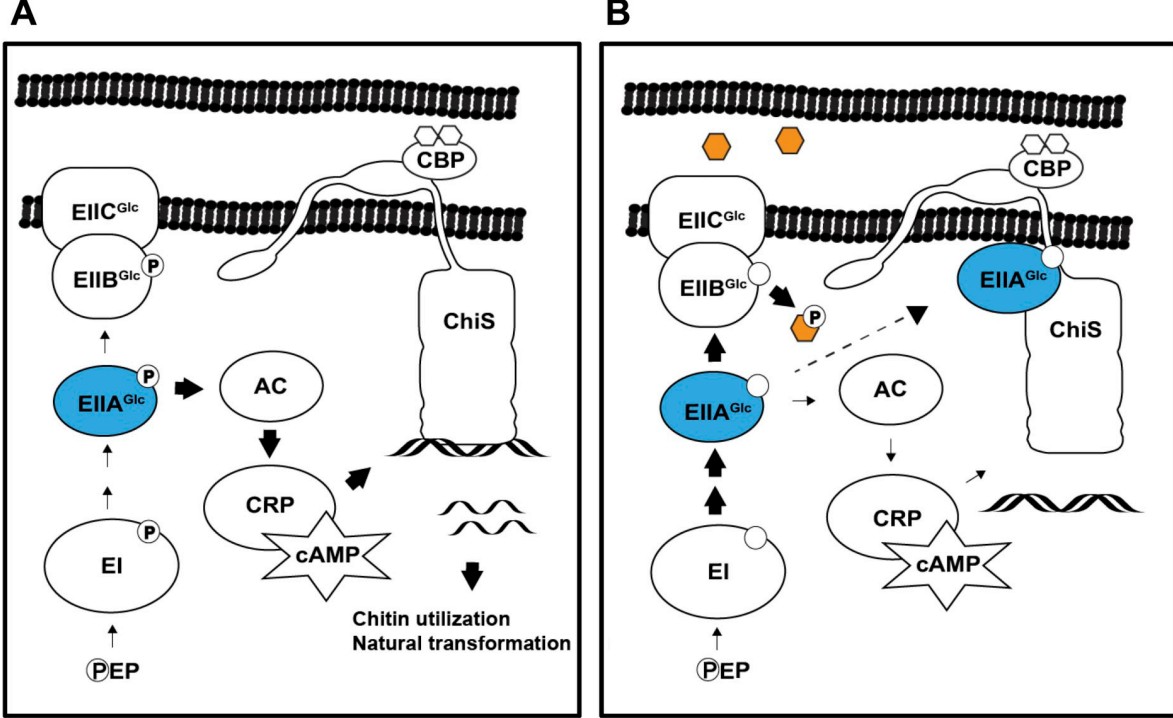

**Fig 7. Model for CCR of the V. cholerae chitin response.** (**A**) In the absence of glucose, phosphate (**P**) accumulates on PTS components due to the absence of a substrate. Thus, EIIA^Glc (blue) is phosphorylated and unable to interact with ChiS. When activated by chitin-bound CBP, ChiS binds DNA and promotes the transcription of genes required for natural transformation and chitin utilization. Phospho-EIIA^Glc also interacts with adenylate cyclase (AC) to strongly induce its activity, which results in the production of high levels of cAMP. This cAMP complexes with the cAMP receptor protein (CRP) to stimulate both ChiS activity and natural transformation. (**B**) In the presence of glucose (orange hexagon), rapid phosphotransfer through the PTS results in dephosphorylation of EIIA^Glc. The absence of phosphorylated EIIA^Glc reduces adenylate cyclase activity and resultant CRP-cAMP activity. However, our results demonstrate that CCR due to a reduction in cAMP production cannot account for repression of ChiS activity. Instead, we found that dephospho-EIIA^Glc interacts with ChiS to inhibit its DNA-binding activity. Thus, ChiS is unable to activate its target genes, resulting in the loss of chitin utilization and natural transformation.

natural competence and illuminate *Vibrio cholerae* regulation of these critical behaviors in response to nutrient conditions.

Given their energetic cost, it is not surprising that both chitin utilization and natural transformation are tightly regulated processes. Multiple regulators have been identified for ChiS alone [8,25,40]. The conservation of a pathway for regulation of these behaviors by PTS sugars raises the question of where *V. cholerae* cells might encounter such sugars while the chitin response is active. In the context of a chitin-associated biofilm, the ability to rapidly inhibit the chitin response to take advantage of favorable carbon sources (perhaps released from decomposing wildlife) might provide a considerable advantage. Alternatively, it is possible that when this pathogen is ingested, the rapid shutdown of the chitin response within the host gut might aid in colonization. While the primary product of chitinolytic activity is the chitin disaccharide chitobiose, a minor amount of GlcNAc, the monosaccharide subunit of chitin, may be produced [50]. PTS-mediated uptake of GlcNAc should also render EIIA^Glc dephosphorylated. Thus, the uptake of GlcNAc liberated via chitinolytic activity may serve as a negative feedback mechanism to inhibit the chitin response [15].

While it is easy to speculate that glucose inhibits chitin utilization to maximize efficiency in carbon metabolism, the link between carbon availability and natural transformation is less intuitive. In *V. cholerae*, natural transformation and chitin utilization are inextricably linked

due to their activation by ChiS. It is possible that a survival advantage is primarily conferred by the ability to regulate chitin utilization, and that the regulation of natural transformation is a by-product. However, this hypothesis is challenged by data from *Haemophilus influenzae*, where natural competence is similarly inhibited by favorable sugars [51]. Furthermore, in *H. influenzae*, *V. cholerae, and Bacillus subtilis*, the induction of competence coincides with a stressful and nutrient-poor environment [5,51,52]. If competence is beneficial to species in stressful environments, the appearance of favorable carbon sources may lessen the need for such a response.

On a mechanistic level, this study also expands our understanding of how CCR regulates natural transformation. Work in *V. cholerae* has attributed CCR of natural transformation solely to the low cAMP levels that result from sugar-induced EIIA$^{Glc}$ dephosphorylation [22]. We observed that while modulation of intracellular cAMP does play a minor role, the EIIA$^{Glc}$-mediated repression of ChiS activity represents the dominant mechanism for CCR of chitin-induced behaviors in *V. cholerae*. This finding differs from CCR of natural transformation in *H. influenzae*, where fructose-mediated repression can be completely relieved by the addition of exogenous cAMP [51]. The additional layer of regulation present in *V. cholerae* is striking, and as discussed above, may have evolved as a response to the system's unique mode of competence induction. The direct regulation by EIIA$^{Glc}$ may strengthen the cell's control over this costly response by rapidly and efficiently preventing its activation in the presence of favorable carbon sources.

This study also adds another member to the extensive EIIA$^{Glc}$ regulon. The ability of EIIA$^{Glc}$ to physically interact with diverse binding partners is poorly understood. Contributing an additional binding partner has the potential to help define requirements for regulation by EIIA$^{Glc}$, thereby expanding our understanding of how this versatile protein controls a number of key biological processes [19,53–56].

ChiS is a membrane-embedded hybrid sensor kinase. When chitin oligomers enter the periplasm, they trigger a conformational change in ChiS that is transmitted from its periplasmic domain through a transmembrane domain [8,24]. This transmembrane domain connects to a flexible linker domain known as a HAMP (present in Histidine kinases, Adenylate cyclases, Methyl-accepting proteins, and Phosphatases) domain, which in turn propagates the signal to the cytoplasmic portion of the protein to activate DNA binding [8,24]. Interestingly, the W388C suppressor mutation is located within the HAMP domain. The I429T suppressor mutation is located directly after the end of the HAMP domain, but before the beginning of the next domain. We hypothesize that these mutations likely confer resistance to repression in one of two ways. First, they may lie at a binding interface between ChiS and EIIA$^{Glc}$, preventing hydrophobic or electrostatic interactions typically required for effective repression. Or second, they may trigger a conformational change in the downstream domains of the protein such that the true binding interface becomes inaccessible to EIIA$^{Glc}$. Interestingly, a single mutation within a HAMP domain has previously been shown to constitutively activate the downstream sensor kinase [57]. The ability of single mutations in the HAMP to drastically alter downstream protein conformation may lend support to the latter hypothesis. Regardless of their mechanism of action, these ChiS suppressor alleles were integral to test the importance of EIIA$^{Glc}$-dependent repression of ChiS activity on CCR of chitin-induced behaviors in *V. cholerae*.

Lastly, our findings also expand our understanding of the mechanisms by which organisms execute CCR. CCR is a widespread mode of gene regulation in diverse bacterial species. Typically, it involves the repression of proteins directly involved in the process being regulated (often permeases or kinases) [12,30,32,56,58]. In Firmicutes, regulation of transcription factors through protein-protein interactions is also extremely common [59]. However, CCR of transcription factors is less established in Gram-negative organisms [18,59]. Interestingly, a recent

study in *Vibrio vulnificus* identified a transcription factor that can be activated by a direct interaction with dephospho-EIIA$^{Glc}$ [37]. This result coupled with our findings may suggest that EIIA$^{Glc}$-dependent regulation of transcription factors is conserved in Gram-negative microbes.

## Methods

### Bacterial strain and culture conditions

Mutant strains were derived from *Vibrio cholerae* E7946 [60]. Unless otherwise indicated, strains were grown in LB Miller broth at 30˚C rolling and on LB Miller agar. When necessary, LB was supplemented with kanamycin at 50 μg/mL, trimethoprim at 10 μg/mL, spectinomycin at 200 μg/mL, chloramphenicol at 1 μg/mL, carbenicillin at 100 μg/mL, or zeocin at 100 μg/mL.

### Strain construction

Mutant constructs were created using splicing by overlap extension (SOE) PCR as previously described [61]. Constructs were introduced into strains using natural transformation, either chitin-induced [62] or genetically induced using an IPTG-inducible *tfoX* plasmid (pMMB67EH-*tfoX*). In all cases, this plasmid was cured before the strains were used in experiments. Constructs were verified by PCR and/or sequencing (all point mutations and in-frame deletions were verified by sequencing). All genes referenced in this study and their corresponding identifiers within the *Vibrio cholerae* genome can be found in **S1 Table**. All strains used in this study can be found in **S2 Table**. Gene deletions were in-frame unless otherwise indicated. For a list of primers used in this study, see **S3 Table**.

### GFP ChiS activity assays

GFP assays were performed almost exactly as previously described [24]. Briefly, strains were streaked from frozen stocks onto LB agar and grown at 30˚C overnight. Single colonies were then picked into LB broth and grown rolling overnight for 17 hours at 30˚C. Overnight cultures were washed and resuspended to an OD$_{600}$ of 3.0 in Instant Ocean (IO; 7g/L, Aquarium Systems). The GFP fluorescence (excitation: 500 nm; emission: 540 nm) of 200 μL of this resuspension was assessed using a BioTek H1M plate reader. Background fluorescence was determined by assaying a non-fluorescent strain in each assay and was subtracted from all samples.

### Chitin-induced ChiS activity assays

Strains were inoculated from frozen stocks into LB broth and grown rolling overnight at 30˚C. Overnight cultures were then washed and resuspended in IO. 10$^8$ cells suspended in 100 μL of IO were then mixed with 1 mL chitin reactions including 150 μL chitin slurry (53 g chitin powder / L IO), 565 μL IO, and either 185 μL cAMP (5 mM final concentration) or an equal volume of water. Reactions were incubated shaking for 48 hours at 30˚C. Cells were then removed from the chitin by vortexing, suspended in Instant Ocean, and placed on a glass coverslip beneath a 0.2% IO gelzan pad. Cells were imaged using an inverted Nikon Ti-2 microscope with a Plan Apo 60x objective lens, FITC (excitation: 470 nm; emission: 535 nm) and mCherry (excitation: 575 nm; emission: 635 nm) filter cubes, a Hamamatsu ORCAFlash 4.0 camera, and Nikon NIS Elements imaging software. Average fluorescence per cell was quantified using Fiji [63] and the MicrobeJ [64] plugin. The background fluorescence was determined by imaging a non-fluorescent strain and subtracted from samples. To account for internal noise in gene expression, reporter fluorescence was normalized to a constitutively

expressed GFP construct. For each biological replicate, the geometric mean was calculated from ~200 individual cells.

### Insoluble chitin growth curves

Strains were inoculated from frozen stocks into LB broth and grown rolling overnight at 30˚C. Overnight cultures were washed and resuspended in M9 minimal media (1X M9 salts + 2 mM $MgSO_4$ + 30 μM $FeSO_4$ + 0.1 mM $CaCl_2$). $10^4$ cells suspended in 10 μL of M9 were then added to 1 mL chitin reactions containing 150 μL chitin slurry, 655 μL M9 minimal media, and either 185 μL cAMP (5 mM final concentration) or an equal volume of water. Growth reactions were grown shaking at 30˚C for 72 hours. Quantitative dilution plating was performed at 0, 24, 48, and 72 hours.

### Transformation assays

Strains were inoculated from frozen stocks into LB broth and grown rolling overnight at 30˚C. Overnight cultures were subcultured and grown to an $OD_{600}$ of 0.5–1.0. Subcultured cells were washed and resuspended in IO, and $10^8$ cells suspended in 100 μL IO were mixed with 1 mL chitin reactions including 150 μL chitin slurry, 565 μL IO, either 185 μL cAMP (5 mM final concentration) or an equal volume of water, and, when applicable, 0.5% final concentration of either glucose, fructose, maltose, or a cocktail of glutamate and proline. Cells were incubated static at 30˚C for ~18 hours. Then, 550 μL of supernatant was removed from the reactions, and 500 ng of purified transforming DNA (tDNA) was mixed with the cells by light vortexing. The tDNA replaced a frame-shifted transposase gene with a chloramphenicol resistance marker (*i.e.*, ΔVC1807::$Cm^R$). DNA and cells were incubated for 5 hours static at 30˚C. Reactions were then outgrown by adding 500 μL LB broth and shaking at 37˚C for 2 hrs. Reactions were then plated for quantitative culture on both plain LB agar (total viable counts) and LB agar + chloramphenicol (1 μg/mL) (transformants). Transformation frequency was calculated by dividing the number of transformants by the total viable counts.

### Colocalization microscopy

Strains were inoculated from frozen stocks into LB broth + 20 μM IPTG to induce the $P_{tac}$-*chiS-msfGFP* constructs and grown rolling overnight at 30˚C. For POLAR experiments, strains were supplemented with 0.2% arabinose to induce $P_{BAD}$-PopZ. Overexpression of ChiS induces a strong cell curving phenotype in *Vibrio cholerae* for reasons that remain unclear. Because this curvature disrupts polar relocalization and downstream analysis, we deleted the *crvA* gene, which is required for cell curvature in *V. cholerae*, for the strain used in this assay [65]. Overnight cultures were washed and resuspended to an $OD_{600}$ of 0.3 in IO and placed on a glass coverslip beneath a 0.2% IO gelzan pad. Cells were imaged using the Nikon Ti-2 microscope discussed above. GFP and mCherry foci were visualized on Fiji [63] and 100 cells were analyzed per replicate for a total of ≥300 cells per sample. Heat maps were generated using Fiji and the MicrobeJ plugin [64].

### Protein co-immunoprecipitation

Strains containing a $P_{tac}$-*chiS-FLAG* allele and natively expressed $EIIA^{Glc}$-mCherry were inoculated from colonies streaked on LB agar into 50 mL flasks of LB broth supplemented with 1 μM IPTG (to induce $P_{tac}$-*chiS-FLAG*). Flasks were grown shaking at 30˚C for 24 hr. Cultures were then crosslinked with 1 mM disuccinimidyl suberate (DSS) for 30 minutes at room temperature shaking before being quenched with 100 mM Tris for 20 minutes at room

temperature shaking. Quenched samples were washed twice with coIP buffer 1 (50 mM Tris-Cl pH 7.4, 150 mM NaCl, 1 mM EDTA) and then stored at -80°C. Frozen pellets were resuspended in 1 mL coIP buffer 2 (50 mM Tris-Cl pH 7.4, 150 mM NaCl, 1 mM EDTA, 10 mM MgCl2, 1% triton, 2% glycerol). Cells were then lysed by the addition of 1x Fastbreak cell lysis reagent (Promega), 1x protease inhibitor cocktail [100x inhibitor cocktail contained the following: 0.07 mg/mL phosphoramidon (Santa Cruz), 0.006 mg/mL bestatin (MPbiomedicals/Fisher Scientific), 1.67 mg/mL AEBSF (Gold Bio), 0.07 mg/mL pepstatin A (DOT Scientific), 0.07 mg/mL E64 (Gold Bio) suspended in DMSO], 250 μg/mL lysozyme, and 500 units/mL benzonase nuclease (Sigma) for 30 mins rocking at room temperature. Lysates were clarified by centrifuging at 10,000 x g for 10 minutes at 4°C. Then, 50 μL of clarified lysate was mixed with 50 μL 2x SDS PAGE sample buffer (200 mM Tris pH 6.8, 25% glycerol, 1.8% SDS, 0.02% Bromophenol Blue, 5% β-mercaptoethanol) to serve as the "protein input" sample. 1 mL of lysate was then added to 100 μL Pierce anti-DYKDDDDK magnetic agarose equilibrated in coIP buffer 2. The lysate was incubated with the agarose for 2hr at 4°C with end-over-end mixing. Supernatant was then removed using a magnetic separation rack and discarded. The agarose was then washed three times with 500 μL coIP buffer 2 for 10 minutes rocking and then briefly washed one additional time in 500 μL coIP buffer 2. Protein was then eluted off the agarose by resuspension in 100 μL of 3X FLAG elution buffer (3 μL of 5 mg/mL Sigma 3X FLAG peptide solution + 97 μL of coIP buffer 2) followed by 30 minutes of rocking at room temperature. Eluant was collected using a magnetic separation rack and was mixed 1:1 with 2x SDS PAGE sample buffer to serve as "protein elution" sample. All samples were then boiled for 10 minutes. For visualizing ChiS-FLAG, 5 μL of input sample or 15 μL of elution sample were separated on a 7.5% SDS PAGE gel. For visualizing EIIA$^{Glc}$-mCherry, 5 μL of input sample or 15 μL of elution sample were separated on a 15% SDS PAGE gel. Following separation, proteins were electrophoretically transferred to a PVDF membrane. Membranes were then blocked for at least 1 hour in 5% milk and then incubated overnight rocking with either rabbit polyclonal α-FLAG (1:1000, Sigma) or rabbit polyclonal α-mCherry (1:10,000, ThermoFisher). Membranes were then washed and incubated for at least 2 hours with α-rabbit horseradish peroxidase-conjugated secondary antibodies (1:10,000). Membranes were then washed and incubated with both primary and secondary antibodies again to maximize signal. Blots were then washed and probed with Pierce ECL Western Blotting Substrate. Blots were imaged with a ProteinSimple FluorChem R system.

### Chromatin immunoprecipitation assays

ChIP assays were performed almost exactly as previously described [24]. Strains were inoculated from frozen stocks into LB broth and grown rolling overnight at 30°C. Overnight cultures were diluted into fresh LB to an $OD_{600}$ of 0.08 and grown for 6 hours at 30°C rolling. Subcultured cells were then crosslinked with 1% paraformaldehyde for 20 minutes at room temperature with shaking before being quenched with a 1.2 molar excess of Tris for 10 minutes at room temperature shaking. Cells were then washed twice with TBS (25 mM Tris HCl, pH 7.5 and 125 mM NaCl), and cell pellets were stored at -80°C overnight. Pellets were then resuspended to an $OD_{600}$ of 50.0 in lysis buffer (1x FastBreak cell lysis reagent (Promega), 50 μg/mL lysozyme, 1% Triton X-100, 1 mM PMSF, and 1x protease inhibitor cocktail; 100x inhibitor cocktail contained the following: 0.07 mg/mL phosphoramidon (Santa Cruz), 0.006 mg/mL bestatin (MPbiomedicals/Fisher Scientific), 1.67 mg/mL AEBSF (Gold Bio), 0.07 mg/mL pepstatin A (DOT Scientific), 0.07 mg/mL E64 (Gold Bio) suspended in DMSO). Lysis reactions were incubated for 20 mins rocking at room temperature and were then sonicated with a Fisherbrand Model 705 Sonic Dismembrator 6 times for 10 seconds at 1% amplitude, resting on

ice for at least 30 seconds between each sonication (yielding sheared DNA fragments of ~200–300 bp). Sonicated lysates were then centrifuged and diluted 1:5 in IP buffer (50 mM HEPES NaOH pH 7.5, 150 mM NaCl, 1 mM EDTA, and 1% Triton X-100). A sample of diluted lysate was collected to permit quantification of input DNA (see below). 1 mL of diluted lysate was then incubated with end-over-end mixing at room temperature with Pierce anti-DYKDDDDK magnetic agarose equilibrated in IP buffer. Supernatant was then removed using a magnetic separation rack. The magnetic agarose was then washed twice with IP Buffer for 1 minute, once with Wash Buffer 1 (50 mM HEPES NaOH pH 7.5, 1 mM EDTA, 1% Triton X-100, 500 mM NaCl, and 0.1% SDS) for 5 minutes, once with Wash Buffer 2 (10 mM Tris HCl pH 8.0, 250 mM LiCl, 1 mM EDTA, 0.5% NP- 40, and 1% Triton X-100) for 5 minutes, and once with Buffer TE (10 mM tris pH 8.0 and 1 mM EDTA) for 1 minute. To elute protein and DNA off the agarose, it was incubated in SDS elution buffer (50 mM Tris HCl pH 7.5, 10 mM EDTA, and 1% SDS) for 30 minutes at 65˚C. The supernatant was then separated from beads using a magnetic separation rack, moved to a new tube, and subjected to digestion with 20 μg Proteinase K in SDS Elution Buffer for 2 hours at 42˚C. Crosslinks were then reversed by incubating samples for 6 hours at 65˚C. Samples were collected following this incubation for quantification of output DNA. DNA within the samples (input and output) was then purified. Input DNA was diluted 1:100 in buffer EB. Quantitative PCR using iTaq Universal SYBR Green Supermix (Bio-Rad) was then used to measure the abundance of the *chb* promoter in the output (following immunoprecipitation) relative to the input (prior to immunoprecipitation). A standard curve generated from genomic DNA diluted in buffer EB permitted quantification of *chb* abundance. Furthermore, *chb* promoter levels were normalized to levels of a gene that is *not* bound by ChiS, *rpoB*, generating a fold enrichment of the *chb* promoter for each sample.

## Western blotting

Strains were inoculated from frozen stocks into LB broth and grown rolling overnight at 30˚C. Overnight cultures were washed and resuspended to an $OD_{600}$ of 100.0 in IO and then mixed 1:1 with 2x SDS PAGE sample buffer (200 mM Tris pH 6.8, 25% glycerol, 1.8% SDS, 0.02% Bromophenol Blue, 5% β-mercaptoethanol). Samples were boiled for 10 minutes. For visualizing ChiS-FLAG, 4 uL of sample were separated on a 7.5% SDS PAGE gel. For visualizing RpoA as a loading control, 2 uL of sample were separated on a 15% SDS PAGE gel. Following separation, proteins were electrophoretically transferred to a PVDF membrane. Membranes were then blocked for at least 1 hour in 5% milk and then incubated overnight rocking with either rabbit polyclonal α-FLAG (1:1000, Sigma) or mouse monoclonal α-RpoA (1:1000, Biolegend). Membranes were then washed and incubated for at least 2 hours with either α-rabbit or α-mouse horseradish peroxidase-conjugated secondary antibodies (1:10,000). Blots were then washed and probed with Pierce™ ECL Western Blotting Substrate. Blots were imaged with a ProteinSimple FluorChem R system.

## Glucose and tryptone growth curves

Strains were inoculated from frozen stocks into LB broth and grown rolling overnight at 30˚C. Overnight cultures were then washed and resuspended in 1X M9 minimal salts + 2 mM $MgSO_4$ + 0.1 mM $CaCl_2$. Cells were inoculated at a starting $OD_{600}$ of 0.001 in 200 μL M9 growth reactions (1X M9 minimal salts + 2 mM $MgSO_4$ + 0.1 mM $CaCl_2$ + 5 μM $FeSO_4$) supplemented with 0.1% glucose or tryptone. The $OD_{600}$ was kinetically monitored every 40 mins in a BioTek H1M plate reader shaking at 30˚C.

## Statistical comparisons

All statistical comparisons were made by one-way ANOVA with Tukey's multiple comparison test. For ratiometric data (*i.e.*, chitin-induced $P_{chb}$ induction assays, growth assays, transformation frequency assays, ChIP assays, and colocalization assays) statistical comparisons were performed on log-transformed data. For summary statistics and a complete list of all statistical comparisons, see **S1 Dataset**.

## Supporting information

**S1 Fig. Dephosphorylated EIIA$^{Glc}$ represses WT but not suppressor ChiS transcriptional activity in the presence or absence of exogenous cAMP. (A)** ChiS activation of the *chb* promoter was assessed in rich medium (in the absence of chitin) using a $P_{chb}$-*GFP* reporter in the indicated strain backgrounds. (**B**) ChiS activation of $P_{chb}$-*mCherry* was assessed in the indicated strains after cells were incubated on chitin for 48 hrs. mCherry signal was normalized to a constitutively expressed GFP construct. All experiments were performed in the presence or absence of 5 mM exogenous cAMP as indicated. All data for samples where exogenous cAMP was added are identical to that present in **Fig 1A and 1B** and are included here for ease of comparison. Results are from at least three independent biological replicates and shown as the mean ± SD. Statistical comparisons were made by one-way ANOVA with Tukey's multiple comparison test. NS, not significant. *** = $p < 0.001$, ** = $p < 0.01$, * = $p < 0.05$.
(TIFF)

**S2 Fig. Dephosphorylated EIIA$^{Glc}$ represses growth on chitin of WT but not suppressor ChiS in the presence or absence of exogenous cAMP. (A)** Final growth yield of the indicated strains after a 72 hr incubation in M9 minimal media containing chitin as the sole carbon source. Reactions were supplemented with 5 mM cAMP as indicated. Results are from at least three independent biological replicates and shown as the mean ± SD. Statistical comparisons were made by one-way ANOVA with Tukey's multiple comparison test. NS, not significant. *** = $p < 0.001$, ** = $p < 0.01$, * = $p < 0.05$. Data for reactions supplemented with cAMP are identical to that presented in **Fig 1C** and are included here for ease of comparison. (**B-C**) Representative growth curves of the indicated strains in M9 minimal media containing chitin as the sole carbon source. Reactions either (**B**) lacked exogenous cAMP or (**C**) were supplemented with 5mM cAMP. Strains containing suppressor alleles are colored (orange for W388C, blue for I429T). Data are representative of at least three independent biological replicates.
(TIFF)

**S3 Fig. Dephosphorylated EIIA$^{Glc}$ represses natural transformation of WT but not suppressor ChiS in the presence or absence of exogenous cAMP.** Chitin-induced natural transformation assays of the indicated strains. Reactions were supplemented with 5 mM exogenous cAMP as indicated. All data for samples where exogenous cAMP was added are identical to that presented in **Fig 1D** and are included here for ease of comparison. Results are from at least three independent biological replicates and shown as the mean ± SD. Statistical comparisons were made by one-way ANOVA with Tukey's multiple comparison test. NS, not significant. *** = $p < 0.001$, ** = $p < 0.01$, * = $p < 0.05$. LOD, limit of detection.
(TIFF)

**S4 Fig. The EIIA$^{Glc}$-mCherry fusion is fully functional for inhibiting ChiS activity.** ChiS activation of the *chb* promoter was assessed in rich medium (in the absence of chitin) using a $P_{chb}$-GFP reporter in the indicated strain backgrounds. Results are from three independent biological replicates and shown as the mean ± SD. Statistical comparisons were made by one-

way ANOVA with Tukey's multiple comparison test. NS, not significant. $^{***} = p < 0.001$, $^{**} = p < 0.01$, $^{*} = p < 0.05$.
(TIFF)

**S5 Fig. CCR is maintained in strains overexpressing ChiS.** ChiS activation of the *chb* promoter was assessed in rich medium (in the absence of chitin) using a P*chb*-GFP reporter in the indicated strain backgrounds. Strains were grown in the present of 1 μM IPTG to induce ChiS-FLAG constructs. Results are from three independent biological replicates and shown as the mean ± SD. Statistical comparisons were made by one-way ANOVA with Tukey's multiple comparison test. Statistical identifiers directly above bars represent comparisons to the parent (teal, first bar). NS, not significant. $^{***} = p < 0.001$, $^{**} = p < 0.01$, $^{*} = p < 0.05$.
(TIFF)

**S6 Fig. ChiS suppressor alleles are expressed at WT levels under CCR conditions.** Western blot analysis for ChiS expression. Strains contained a ΔEI mutation and the indicated FLAG-tagged ChiS allele. A strain lacking a FLAG-tagged ChiS allele was also included as a negative control (No FLAG). Blots were developed with primary antibodies against FLAG or RpoA as a loading control as indicated. Data are representative of three independent biological replicates.
(TIFF)

**S7 Fig. The ChiS^W388C allele confers resistance to glucose-induced CCR of natural transformation in the presence or absence of exogenous cAMP.** Chitin-induced natural transformation assays of the indicated strains. (**A**) Transformation reactions were supplemented with 0.5% glucose (Glu), fructose (Fru), maltose (Malt), or a mixture of glutamate and proline (E P) as indicated. (**B**) Transformation reactions were supplemented with 0.5% glucose as indicated. Transformation reactions were supplemented with 5 mM cAMP as indicated. Results are from three independent biological replicates and shown as the mean ± SD. Statistical comparisons were made by one-way ANOVA with Tukey's multiple comparison test. NS, not significant. $^{***} = p < 0.001$, $^{**} = p < 0.01$, $^{*} = p < 0.05$. LOD, limit of detection. In **A**, statistics directly above data represent comparison to the parent in equivalent cAMP condition (teal, first two bars and blue, fifth and sixth bars). Data for reactions supplemented with exogenous cAMP in **B** are identical to that presented in **Fig 5** and are included here for ease of comparison.
(TIFF)

**S8 Fig. The ΔPGI mutation prevents growth on glucose but not amino acids.** Representative growth curves of the indicated strains in M9 minimal medium containing 0.1% (**A**) glucose or (**B**) tryptone as the sole carbon source. Data are representative of three independent biological replicates.
(TIFF)

**S9 Fig. *CCR of ChiS is conserved in diverse* V. cholerae *strains as well as in* Vibrio campbellii.** ChiS activation of P*chb*-mCherry was assessed in the indicated strains after cells were incubated on chitin for 48 hrs (*V. cholerae* strains) or 96 hrs (*V. campbellii* DS40M4) in the presence of 5 mM cAMP. mCherry signal was normalized to a constitutively expressed GFP construct. Each strain is shown in a different color, with *V. cholerae* E7946 (the parent for all other strains used in this study) shown in teal, the non-toxigenic environmental *V. cholerae* isolate CR03-424 shown in pink, the toxigenic *V. cholerae* isolate C6706 shown in orange, the toxigenic *V. cholerae* isolate A1552 shown in blue, and the *V. campbellii* DS40M4 strain shown in yellow. Data are from at least three independent biological replicates and are shown as the mean ± SD. Statistical comparisons were made by one-way ANOVA with Tukey's multiple comparison test. NS, not significant. $^{***} = p < 0.001$, $^{**} = p < 0.01$, $^{*} = p < 0.05$. Statistical

identifiers directly above bars represent comparisons to the parent of the same strain (first column in each color-coded group).
(TIFF)

**S1 Table. Genes referenced in this study.**
(PDF)

**S2 Table. Strains used in this study.**
(PDF)

**S3 Table. Primers used in this study.**
(PDF)

**S1 Dataset. Summary statistics and statistical comparisons for all data in this study.**
(XLSX)

## Acknowledgments

We would like to thank Dan Kearns, Jake McKinlay, Julia van Kessel, Clay Fuqua, Jen Greenwich, and Courtney Ellison for helpful discussions. We would also like to thank Julia van Kessel for *V. campbellii* DS40M4 and Clay Fuqua for reagents.

## Author Contributions

**Conceptualization:** Virginia E. Green, Ankur B. Dalia.

**Investigation:** Virginia E. Green, Catherine A. Klancher, Shouji Yamamoto.

**Writing – original draft:** Virginia E. Green, Ankur B. Dalia.

**Writing – review & editing:** Virginia E. Green, Catherine A. Klancher, Shouji Yamamoto, Ankur B. Dalia.

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
