## [Decision Letter · Decision Letter 0]

2 Nov 2022

Dear Dr Dalia,

Thank you very much for submitting your Research Article entitled 'The molecular mechanism for carbon catabolite repression of the chitin response in Vibrio cholerae' to PLOS Genetics.

The manuscript was fully evaluated at the editorial level and by independent peer reviewers. The reviewers appreciated the attention to an important problem, but raised some substantial concerns about the current manuscript. Based on the reviews, we will not be able to accept this version of the manuscript, but we would be willing to review a much-revised version. We cannot, of course, promise publication at that time.

If you decide to revise the manuscript for further consideration at PLOS Genetics, please aim to resubmit within the next 60 days, unless it will take extra time to address the concerns of the reviewers, in which case we would appreciate an expected resubmission date by email to plosgenetics@plos.org.

We are sorry that we cannot be more positive about your manuscript at this stage. Please do not hesitate to contact us if you have any concerns or questions.

Yours sincerely,

Jan-Willem Veening, Ph.D.

Academic Editor

PLOS Genetics

Lotte Søgaard-Andersen

Section Editor

PLOS Genetics

Editor notes:

While the referees feel this work is potentially very interesting, the current data does not fully support the claims made in the manuscript and the complete molecular mechanism responsible for carbon catabolite repression of chitin-dependent responses in Vibrio cholerae is not fully uncovered. Several experiments are proposed by Reviewer 2 that would require significant amounts of new work, such as showing supporting experimental evidence for a direct interaction between ChiS and EIIA, as well as showing the conservation of the response in other Vibrio strains. I agree with this assessment. Nevertheless, if you think these experiments are doable, I invite you to resubmit a revised version, but for time efficiency you might want to consider a more specialized journal.

Reviewer's Responses to Questions

**Comments to the Authors:**

Reviewer #1: This is nicely detailed genetics work from the Dalia group that demonstrates the molecular details behind carbon catabolite repression on the utilization of chitin polymer and competence for DNA uptake in Vibrio cholerae. Initiated by previous studies, they here dissect the proteins (EIIA-Glc and ChiS) involved in connecting CCR and chitin-induced regulation. The molecular details are appreciated and the numerous alternative approaches demonstrate the role of specific protein-protein interaction. The experiments seem well-controlled and properly explained. Only minor comments.

The term evolution (line 18, 28, 44, 310) is too strong here, as the exchange of genetic material does not equal evolution (=mutations/genetic variance + selection/adaptation). The authors should rather refer to genetic exchange or HGT, but not evolution directly.

Line 46: Another disadvantage of excreted enzymes, like chitinase of Vibrio is the exploitability, which could be mentioned in this sentence (see https://doi.org/10.1016/j.cub.2013.10.030).

Line 78: introduce what “dephospho-” means. This becomes clear at the results section, but should be clear also here for the readers.

Line 387: include respective filter excitation and emission wavelength ranges

Statistics are lacking in the figures, which should be added for all figures, but becomes especially relevant for Fig 1C, where some of the differences are not as huge as in other panels. Nevertheless, even if the differences are high, significance should be added for all. Also, the data points would be preferable to include for each bar graph, which is becoming a standard for the field (see example: https://www.nature.com/articles/s41551-017-0079).

Reviewer #2: In this manuscript, the authors delineate a novel mechanisms by which carbon catabolite repression of chitin-inducible gene expression might occur in Vibrio. The work is interesting in that it points to a causal involvement of EIIA-Glc. At the same time, the work is firmly rooted in previously published data (ref 22 and others), and lacks some of the data that are needed to support the claim of a direct interaction (as posed in Figure 7). Overall, I feel that the manuscript would benefit from further experimentation as well as a critical description of the results.

Major points:

1. The authors do not address whether this is a conserved response. With large (serogroup) variety, it would be important to know whether the observed effects are broadly conserved or limited to a specific strain of V. cholera. This type of broad impact is what I would expect for a PGEN manuscript.

2. The manuscript presents a strong focus on EIIAGlc, and provides limited room for alternative hypotheses. Most data could also be explained by an indirect effect of EIIAGlc on ChiS (whether likely or not). I think the data need to be more carefully discussed in this view. To claim a direct interaction between ChiS and EIIAGlc, the hypothesis should be supported by biochemical work (crosslinking in Vibrio or in vitro, pulldowns using the two proteins, DNA binding assays in vitro when possible) showing a direct interaction [the POLAR assay can at best be interpreted as a likely interaction in a heterologous host and may depend on other proteins in such host]; also, if a specific interaction exists between EIIAGlc and ChiS, it should be possible to find suppressors in EIIAGlc as well to support this notion. Modeling (e.g. Alphafold/Alphafold-Multimer) might also provide further support. Does the phosphorylation state of EIIAGlc affect the phosphorylation state of the other components (as one would expect in a relay) and if so, how does this factor into the results? In the absence of more concrete data, the claim of a direct interaction (e.g. L327) are insufficiently supported.

1. Figures are not consistent: why are y-axis for the same variable (Fluorescence in AU, though beit mCherry and GFP in different panels) sometimes given on a linear scale and sometimes on a log scale? This complicates interpretation of the data. Similarly, why is data sometimes represented by single datapoints (e.g. Fig 1B) and sometimes in a bar chart? Some of the graphs (e.g. S2 are difficult to read and might benefit from additional color coding for specific groups. For traceability, I suggest including relevant strain names in the legend of Figures (see column Reference in manuscript column). Table S2: why do strains have double identifiers? Are these different strains? If they are the same, why list two identifiers?

3. A key to incorporating the present findings in our thinking of chitin-inducible gene expression is the model as presented in Figure 7. Unfortunately, in its present form this figure is not very clear. I suggest concrete steps to improve the figure: a) a more schematic approach in which elements are aligned, less angled and where possible further simplified in shape; b) a more user-friendly color scheme (including only using color for those elements that are crucial to the model); c) a clear description of all elements in the Legend (at present for instance Glc and Glc~P are not described).

Minor points:

2. Introduction: how relevant is CCR in the natural habitat of Vibrio. Are Glc levels sufficient for repression expected? I think it would be relevant to make a statement to this effect in the Introduction.

3. Authors state that EIIAGlc is also responsible for GlcNAc import (L69) and that GlcNac is a constituent of chitin (L37). How does this fit into the authors statement of a “switch” between glucose and chitin utilization (L47-48)?

4. Fig 1B: what is the level of fluorescence in this assay for a strain lacking Pchb-mCherry (or ideally having a promoterless mCherry)? The authors claim “full repression by EI deletion”(L128) but levels are significantly above that of the chiS mutant. Please rephrase.

5. L140: what data in Fig 1B supports this claim?

6. Figure 2: lacks a control of a chiS mutant (or a FLAG-less allele). This is a necessary control for the specificity of the ChIP data, as well as an interpretation of the enrichment still observed in the CBP/EI. Legend and Figure should indicate that these are data from a FLAG-tagged ChiS (the authors state it is fully functional, but data is NOT derived from a non-tagged construct).

7. Fluorescence colocalization studies (Fig 3A and Fig 5A) are not very convincing – in particular, EIIAGlc foci are rather poor; perhaps this I the rendering of the figure and it can be improved by better pictures. But a higher resolution method or FRET/BRET-based approach would be preferential to demonstrate colocalization more conclusively. Based on this data, I do not agree with the statement that in L253.

**Have all data underlying the figures and results presented in the manuscript been provided?**

Reviewer #1: Yes

Reviewer #2: **No: **To my knowledge, the raw data underlying the graphs is not made available (see also point about data presentation)

PLOS authors have the option to publish the peer review history of their article (what does this mean?). If published, this will include your full peer review and any attached files.

Reviewer #1: **Yes: **Akos T Kovacs

Reviewer #2: **Yes: **Wiep Klaas Smits

---

## [Editor Report · Decision Letter 1]

30 Apr 2023

Dear Dr Dalia,

We are pleased to inform you that your manuscript entitled "The molecular mechanism for carbon catabolite repression of the chitin response in Vibrio cholerae" has been editorially accepted for publication in PLOS Genetics. Congratulations!

Yours sincerely,

Jan-Willem Veening, Ph.D.

Academic Editor

PLOS Genetics

Lotte Søgaard-Andersen

Section Editor

PLOS Genetics

Comments from the reviewers (if applicable):

**Data Deposition**

http://datadryad.org/submit?journalID=pgenetics&manu=PGENETICS-D-22-01115R1

**Press Queries**

---

## [Editor Report · Acceptance letter]

9 May 2023

PGENETICS-D-22-01115R1 

The molecular mechanism for carbon catabolite repression of the chitin response in Vibrio cholerae 

Dear Dr Dalia, 

We are pleased to inform you that your manuscript entitled "The molecular mechanism for carbon catabolite repression of the chitin response in Vibrio cholerae" has been formally accepted for publication in PLOS Genetics! Your manuscript is now with our production department and you will be notified of the publication date in due course.

With kind regards,

Zsofia Freund

PLOS Genetics

On behalf of:
